

# Emerging framework for attack detection in cyber-physical systems using heuristic-based optimization algorithm

Manal Abdullah Alohali[1], Muna Elsadig[1], Anwer Mustafa Hilal[2] and Abdulwahed Mutwakel[3]

[1] Department of Information Systems, College of Computer and Information Sciences, Princess Nourah bint Abdulrahman University, Riyadh, Saudi Arabia
[2] Department of Computer and Self Development, Prince Sattam bin Abdulaziz University, Saudi Arabia, Saudi Arabia, Saudi Arabia
[3] Department of Information Systems, Prince Sattam bin Abdulaziz University, Saudi Arabia, Saudi Arabia, Saudi Arabia

Corresponding author
Anwer Mustafa Hilal,
a.hilal@psau.edu.sa

## ABSTRACT

In recent days, cyber-physical systems (CPS) have become a new wave generation of human life, exploiting various smart and intelligent uses of automotive systems. In these systems, information is shared through networks, and data is collected from multiple sensor devices. This network has sophisticated control, wireless communication, and high-speed computation. These features are commonly available in CPS, allowing multi-users to access and share information through the network *via* remote access. Therefore, protecting resources and sensitive information in the network is essential. Many research works have been developed for detecting insecure networks and attacks in the network. This article introduces a framework, namely Deep Bagging Convolutional Neural Network with Heuristic Multiswarm Ant Colony Optimization (DCNN-HMACO), designed to enhance the secure transmission of information, improve efficiency, and provide convenience in Cyber-Physical Systems (CPS). The proposed framework aims to detect attacks in CPS effectively. Compared to existing methods, the DCNN-HMACO framework significantly improves attack detection rates and enhances overall system protection. While the accuracy rates of CNN and FCM are reported as 72.12% and 79.56% respectively, our proposed framework achieves a remarkable accuracy rate of 92.14%.

## INTRODUCTION

Detecting such attacks has become a complicated issue for securing network systems from attacks. Attackers may penetrate the network and transmit the wrong information when transmitting information from various sensor devices. Moreover, attackers can penetrate the network and hack or steal the credentials or sensitive information, resulting in a loss. Therefore, detecting the attack in the network plays a vital role (*Latif et al., 2020*). Using a cyber-physical system (CPS) is the inception of internet connectivity with various smart devices, technologies, robots, *etc.*, (*Duo, Zhou & Abusorrah, 2022*). In recent years, the application of CPS has increased, such as in the transportation system, distributors of

water and gas, grids, and health care system. Since CPS can be controlled and activated by the function of a wireless communication network with sensor devices, many intruders can easily penetrate through wireless communication networks. Therefore, standardized security measures, *i.e.*, a cyber security process, are required in the CPS to detect and prevent attacks on the network system. This cyber security process is based on the artificial intelligence concept for handling heterogeneous data. The effective detection of attacks in heterogeneous data is carried out by various types of machine learning algorithms (*Yang et al., 2020*).

Transmission of data from various terminal devices *via* the network so that multi-users can access it. It may create the insecurity of data in the distributed system. *Farivar et al. (2019)* presented the detection of attacks using artificial intelligence-based IoT, which will detect malicious attacks in the CPS. *Shuwandy et al. (2020)* describe the CPS of various devices with a cloud storage environment. Many research works have been developed to detect insecure networks and attacks in the network. Inefficiency and the high detection error rate are issues in the existing algorithms. In our proposed approach, a deep convolutional neural network-based bagging concept with an optimized algorithm of Heuristic Multiswarm Ant colony Optimization (DCNN-HMACO) is implemented.

The main contribution of this work is:

1. Implementing network attack detection in a cyber-physical system by applying the pre-processing concept.
2. Extracting the features in the dataset using a linear discriminant algorithm.
3. Enhancing the efficient detection of attacks using deep convolution neural network-based bagging concept with DCNN-HMACO.

The article has been organized as follows: literature is reviewed in "Review of Literature", whereas the methodology of attack detection in a cyber-physical system environment using DCNN-HMACO is explained in "Methodology", The experiment results are discussed in "Results and Discussion", and "Conclusion" concludes the article with future directions.

## REVIEW OF LITERATURE

To detect the malicious attack in the network, prevent the attack on the system and resources, and preserve sensitive information from cybercriminal activities of the network, various approaches are implemented (*Disha & Waheed, 2022*). Cyber-physical systems (CPS) have been implemented in connecting different sensor-based devices, IoT-based devices, and robots where information is collected from multiple terminal devices in heterogeneous data (*Deloglos, Elks & Tantawy, 2020*; *Li et al., 2022*). The KDD 99 and NSLKDD 99 datasets identify cyber-attacks in the cyber-physical system (*Thomas & Pavithran, 2018*; *Li et al., 2021*). In the distributed system, information sharing from various terminal devices and machine learning algorithms is applied for attack detection in cyber-physical systems. A gated recurrent unit with deep neural network (DNN) detects

attacks and deep convolutional neural network (DCNN) in identifying DDoS attacks (*Li et al., 2020*; *Hussain et al., 2020*; *Labonne, 2020*).

*Alazzam, Sharieh & Sabri (2020)* described the detection of the Intrusion Detection System (IDS) based on a binary classifier algorithm of random forest and the optimization technique of Pigeon Inspired Optimizer (PIO) for implementing the feature reduction. *Lee, Pak & Lee (2020)* also described the behavior of the optimization algorithm of Pigeon Inspired Optimizer for detecting attacks in the network. *Meftah, Rachidi & Assem (2019)*, *Zheng et al. (2022)*, *Cao et al. (2020)* proposed a framework for detecting IDS with two stages. The first stage uses binary classification and, based on the second stage, detects the attack in the network. For the detection of an attack, it used multiclass classifiers. To improve the efficient detection of attacks, random forest for the feature selection and training of the network with support vector machine (SVM), logistic regression, and gradient boost machine were applied.

DeepMIH likely introduces a novel approach to image hiding using deep invertible networks. Deep invertible networks are a class of deep learning models designed to be both forward and inverse (backward) invertible (*Guan et al., 2022*). Image registration is a fundamental task in computer vision and medical imaging, which involves aligning two or more images of the same scene or object taken from different perspectives, sensors, or modalities (*Deng et al., 2023*; *Qiao, Li & Kong, 2023*). By incorporating these techniques into the fuzzing process, the authors aim to enhance the effectiveness of vulnerability discovery in smart contracts, potentially leading to more robust and secure blockchain applications (*Liu et al., 2023b*). The article addresses the problem of identifying performance anomalies in cloud environments that experience fluctuations or variations (*Song et al., 2023*). The article proposes an innovative method that combines differential privacy with consensus control algorithms to enable safe cooperation and competition in multi-agent systems (*Ma & Hu, 2022*). The article presents a technique that models relation paths for knowledge graph completion. The approach likely involves designing a model that effectively captures and processes relation paths between entities in the knowledge graph (*Shen et al., 2020*; *Lu et al., 2023*). The proposed system probably incorporates techniques to handle the unique challenges of short texts, such as limited context and noisy language (*Liu et al., 2023a*). The research topic likely focuses on how mobile IoT devices can be combined with data physical fusion technology to create more intelligent and efficient systems (*Lv & Song, 2019*). The research or study likely proposes a methodology or algorithm to optimize the deployment of wireless sensor nodes in industrial environments while prioritizing security considerations (*Cao et al., 2019*).

They used the UNSW-NB 15 dataset. *Injadat et al. (2020)*, *Guo et al. (2022)* presented a machine learning-based algorithm of KNN and random forest using the framework of NIDS. Table 1 shows the survey on the existing algorithm in IDS.

## METHODOLOGY

The emerging framework of attack detection in the cyber-physical system is composed of smart terminals (nodes), links between terminals (edges), and attack vector values with

**Table 1 Survey on IDS.**

| Author name | Technique used | Dataset | Limitations |
|---|---|---|---|
| Moustafa (2021) | Random forest | Network TON_IoT | Only one dataset is tested. Also, random forest does not read complex patterns. |
| Gu & Lu (2021) | Naïve Bayes, SVM. | UNSW-NB 15, CICIDS2017, NSL-KDD, and Kyoto 2006 | Naïve Bayes is sensitive to the quality and relevance of features used for classification. The classifier's performance may be negatively affected if essential features are missing or irrelevant features are included. SVM requires selecting appropriate parameters, such as the kernel type and regularization parameter (C). Different choices of these parameters can significantly impact the performance of the SVM classifier. |
| Belgrana et al. (2021) | Radial basis function (RBF), CNN | NSL-KDD | RBF networks are susceptible to overfitting, especially when the number of hidden units is large compared to the available training data. In this work, only one dataset is used. |
| Kasongo & Sun (2020) | XGBoost algorithm, SVM, Logistic regression, KNN, DT, | UNSW-NB 15 | XGBoost can be computationally demanding, mainly when dealing with large-scale CPS datasets. LR can limit its ability to capture complex relationships and nonlinear patterns in CPS attack detection, potentially leading to lower accuracy. KNN involves comparing the distances between instances, which can be computationally expensive for large CPS datasets. |
| Tama & Rhee (2019) | GBM | NSL KDD, UNSW-NB 15 and GPRS | Sensitivity to class imbalance |
| Jing & Chen (2019) | SVM | UNSW-NB 15 | Computed only for one dataset whereas proposed done for two datasets. The performance will be less than the hybrid deep learning model. |
| Aboueata et al. (2019) | SVM | UNSW-NB 15 | Computed only for one dataset, whereas proposed done for two datasets. The performance will be less than the hybrid deep learning model. |
| Tama & Rhee (2019) | GBM | UNSW-NB 15 | Computed only for one dataset, whereas proposed done for two datasets. The performance will be less than the hybrid deep learning model. |

entry points. Smart terminal represents the devices or machines embedded with software *via* the internet. The link between the terminals is used for the transmission of information. During the transmission of information, attackers may attack the terminals and steal the information. Figure 1 demonstrates the various attacks in the IoT-based CPS.

## Types of attacks in the IoT-based CPS
### MiTM

MiTM (Man-in-the-Middle) is a hijack attack that silently monitors the observation of transmission of data between two smart terminal devices. It does not affect the data in the network, but it disrupts the privacy of information. MiTM acts as eavesdropping and poses an extreme threat to the cyber security of hypertext transfer protocol (HTTP) in transferring data. During its transmission of information, this type of attacker can modify the original data by inserting false information into it.

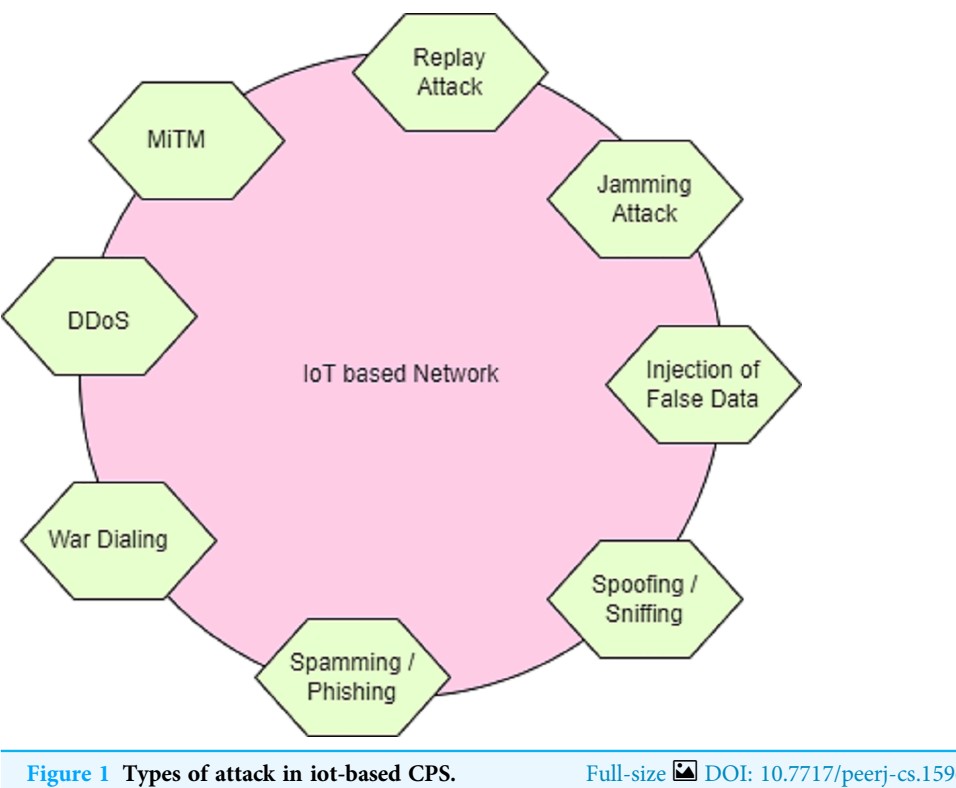

**Figure 1  Types of attack in iot-based CPS.**

### DDoS

A distributed denial of service (DDoS) is a distributed system that generates unwanted traffic in the network and is created by a malicious user. Legitimate users are unaware of the unwanted traffic when transmitting a huge volume of data. Therefore, legitimate users have to wait for a long time to transmit the information. DDoS creates a server down and catastrophe environment in the network.

### Phishing

Phising is when attackers steal legitimate users' identity details by convincing them *via* social media platforms. By using stolen user credentials, attackers steal sensitive information in the network.

### Spamming

The technique of stealing legitimate users' information through social media platforms by email, pop-up messages, advertisements, *etc.* is called spamming. Through spam messages, viruses or worms enter the system and steal the information and upload it to the remote server.

### Jamming

In the network, the jamming device targets the cellular data and wireless communication network and it jams the legitimate user's device. This results in harming the access to the user's services in the network.

### Sniffing & spoofing

In sniffing, the attackers concentrate on the network's data link layer and put a sniffer packet then the user captures that packet, and their network communication gets hacked. In the case of spoofing, the attacker acts like a legitimate user and steals all credential details of the user, and accesses their network.

### Injection of false data

The attacker transmits false information about data packets in the company's network. As a result, it creates negative pricing, the wrong billing process, a downfall in the economy, and a loss of revenue for the company.

### Replay attack

In the system, the attacker employs the fatal attack, which allows the replaying of the attacking process and generates the rebooting of the system, which stores its function of the system.

### War dialing

In the war dialing process, the attacker disconnects the communication link of the user. By applying for the freeware program, it randomly dials the phone number and establishes the connection *via* modem. In the communication system, the attacker detects the loophole and establishes a new connection, and steals the sensitive information of the user.

## Attack detection

This article proposed a framework to detect the attack in the smart terminals of the CPS using a deep bagging convolution neural network with Heuristic Multiswarm Ant colony Optimization (DCNN-HMACO). The framework of DCNN-HMACO is given in Fig. 2.

The proposed framework has four modules, namely, Data Collection, Pre-Processing, Feature Extraction, and Attack Detection using deep bagging convolution neural network with DCNN-HMACO.

### Data collection

The attack detection in the cyber-physical system using DCNN-HMACO and the data set used in this work are UNSW-NB15, and TON_IoT Train_Test Network.

### Pre-processing

In the detection of attacks in the cyber-physical system, data is collected from various smart terminal devices. The collected raw data may generate false detection of attacks in the network. Therefore, pre-processing is required. Figure 3 shows the pre-processing of CPS in DCNN-HMACO.

The pre-processing module contains an implementation of feature scaling using a normalization technique, one hot encoding, replacement of missing values, and removal of redundancy.

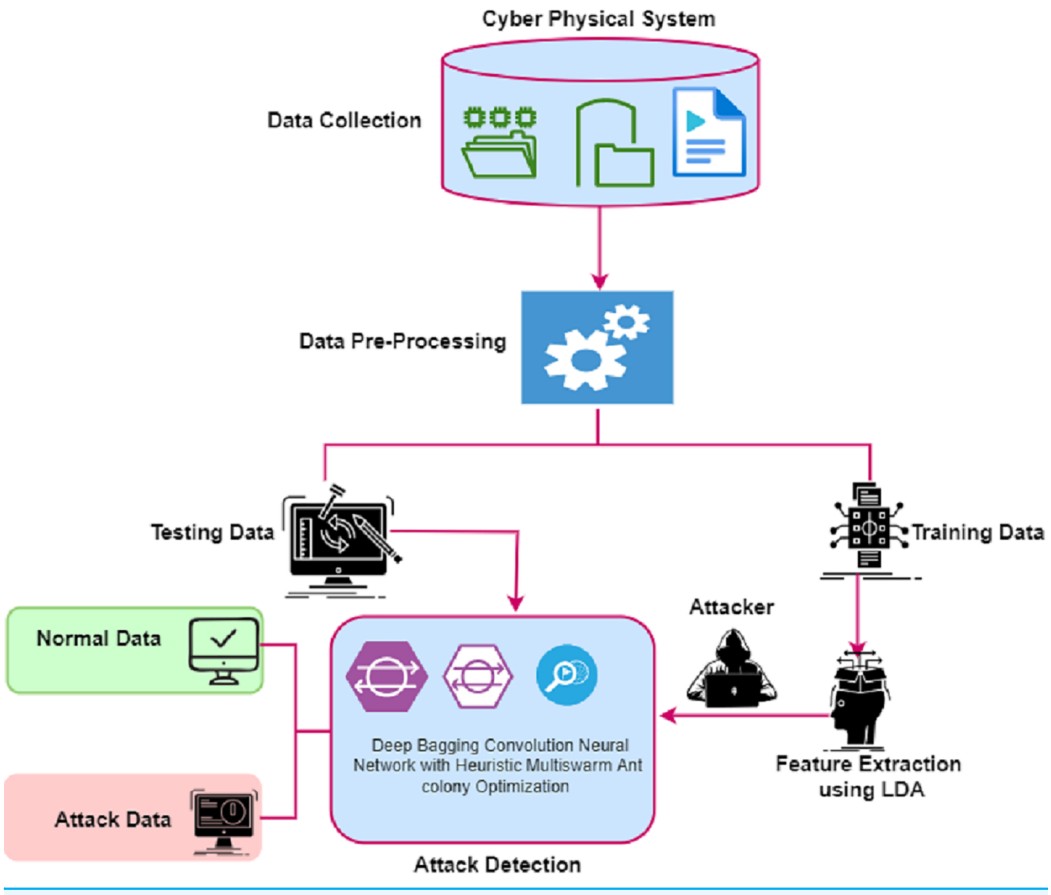

**Figure 2 Framework of the proposed DCNN-HMACO.**

### Feature scaling

The data collected from various smart terminals has various characteristics. Therefore, handling these collected raw data will mislead the detection of the network properly. This article uses the feature scaling process of normalization. It normalized the range of values in the dataset by using Min–Max scaling. The range of normalization is between 0 and 1.

$$fea' = (fea - min(fea))/(max(fea) - min(fea)) \qquad (1)$$

Here $fea'$ is the normalized features in the dataset, $fea$ is the original features in the dataset,

$Min$ and $max$ are values of features in the dataset.

### Response coding based on categorical feature

For encoding the categorical features of the dataset, response coding is used which classifies the features based on their category and represents the probability ($prob$) of data instances in the category. This can be implemented by:

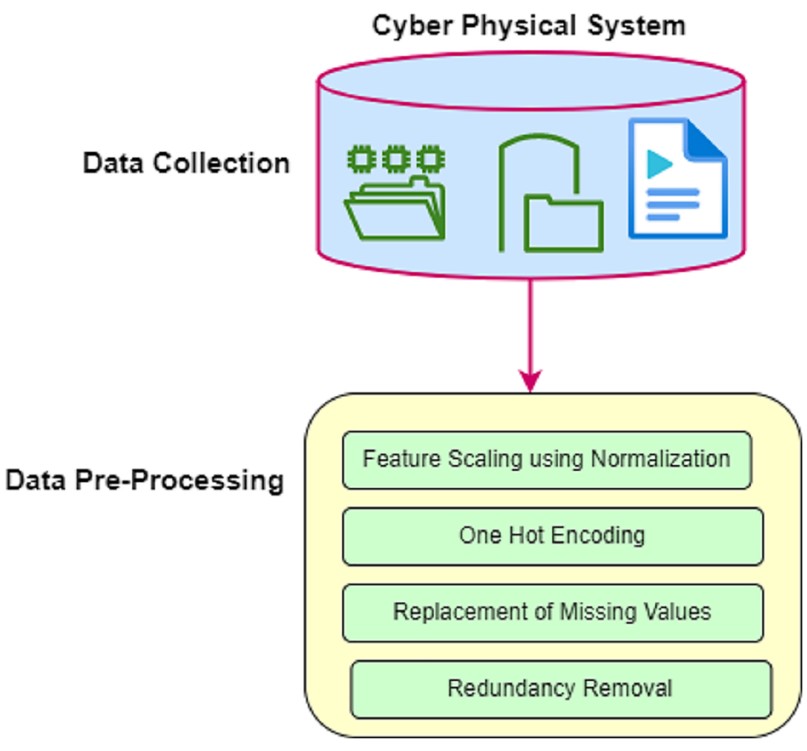

**Figure 3 Data pre-processing.**

$$Prob(F) = \frac{Prob(F \cap X)}{Prob(F)} \tag{2}$$

Here, $X$ denotes the class $F$ F denotes the category feature.

### *Replacement of missing values*

Rearranging the features of the dataset in ascending order and evaluating the median value. Replacing all missing values based on a median value.

### *Redundancy removal*

Accurate and efficient detection of attacks in the network is achieved by eliminating the irrelevant features of the dataset.

### *Feature extraction*

Linear discriminant analysis (LDA) is used to extract features in the data set. Sample data were collected and classified into $p$ pattern classes, data in the $i^{th}$ class has samples of $\sum_{i=1}^{r} y$. The total samples were collected and defined as $a_j$. Choose the $j^{th}$ sample of $i^{th}$ class in $n^{th}$ column vector. Evaluate the projection vector which minimizes the distance between the samples in the data class. LDA is used to construct the projection vector of:

$$n = argarg \frac{n^s V_c n}{n^s V_z n} \tag{3}$$

Here $V_c$ and $V_z$ are scatter matrices in and between classes. These scatter matrices are evaluated by using:

$$V_c = \frac{1}{b}\sum_{i=1}^{x} b_i(w_i - w)(w_i - w)^s \tag{4}$$

$$V_z = \frac{1}{b}\sum_{i=1}^{x}\sum_{j=1}^{b_j}\left(d_j^i - w_i\right)\left(d_j^i - w_i\right)^s \tag{5}$$

$$n = argarg\ n^s(V_z - \lambda V_n)n \tag{6}$$

Here $n$ is the column vector of sample $i$. $b$ is the constant value of magnitude and it is a positive value. After implementing the Eq. (6), $f$ features are selected from the dataset and form the projection of the feature vector.

## Attack detection using DCNN-HMACO (proposed)

### Deep CNN

First, the DCNN with HMACO is applied for network attack detection, then LDA is used for the extraction of features from the dataset. The selection of features is implemented using DCNN. Traditionally, a deep convolutional neural network consists of the input layer, convolution layer, pooling layer, full connection layer, and output layer (*Wang, 2020*; *Yao et al., 2023*). In the convolution layer, features are extracted from the data in the dataset, and the outcome is transmitted into the lower layer. In this layer, the activation function is applied. In the pooling layer, sub-sampling reduces data from the convolution layer. In the full connection layer, nodes are connected with all nodes of the previous layer. The output layer is used to detect attacks in the network by using the softmax function. In this proposed work, attack detection in the network Deep CNN has been enhanced with the bagging operation. This bagging operation has replaced the output layer of traditional-based CNN. The outcome of the convolution layer and pooling layer are fed as input to the bagging concept of the ensemble-based classifier. The detection of attack in the network is based on the maximum voting of the ensemble classifier. The architecture of Deep Bagging CNN is shown in Fig. 4 where the bagging method is used in the training of the network model.

Let us consider $N$ as the number of network layers. Assume that the kernel size of the convolution network is $k$. The dimension of the kernel matrix is defined as $D$. The procedures involved in the DBCNN are given as follows:

**Input:** Data set $D$, Number of features $fn$.

**Step 1:** Initialize parameters of weight $wt$, bias $bi$ and the maximum number iteration $iter$ and the threshold $\varepsilon$.

**Step 2:** In the training phase, the network model is trained by forward propagation, and backward propagation along with updating the weight between the layers, and bias value.

**Step 3:** The **forward propagation phase**, train the data set with given as input and its output is evaluated by:

**Step 4:** *For convl = 2 to N − 1*
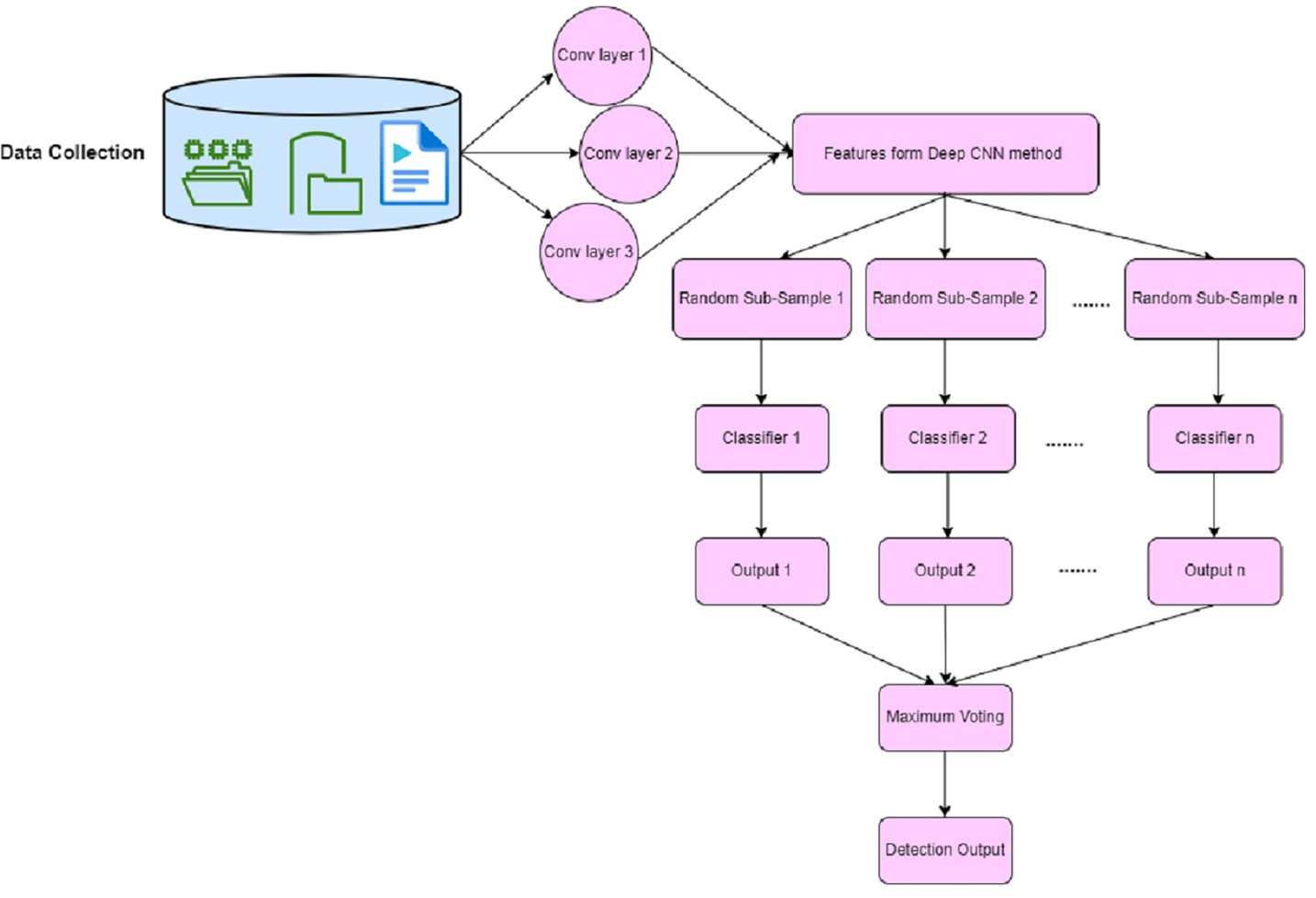

**Figure 4 Architecture of deep bagging CNN.**

**a)** If *convl* is the convolution layer, then the missing data after filling ($a^{convl}$) is represented in Eq. (7)

$$a^{convl} = ReLU\left(z^{convl}\right) = ReLU\left(a^{convl} \times wt^{convl} + bi^{convl}\right) \tag{7}$$

**b)** If *convl* is the pool layer then,

$$a^{convl} = pool(a^{convl-1}) \tag{8}$$

**c)** If *convl* is a full connection layer then,

$$a^{convl} = \sigma\left(z^{convl}\right) = \sigma(wt^{convl} a^{convl-1} + bi^{convl}) \tag{9}$$

**Step 5:** End for
**Step 6:** The output layer of *ol* is represented in Eq. (10)

$$a^{L} = softmax\left(z^{oL}\right) = softmax(wt^{oL} a^{oL-1} + bi^{oL}) \tag{10}$$

**Step 7:** In the **Backward propagation phase**, the error is evaluated between actual output with its corresponding output.

**Step 8:** *For convl = 2 to N − 1*

**a)** If *convl* is the fully connection layer then,

$$\delta^{i,convl} = (wt^{convl+1})^T . \delta^{i,convl+1} \ominus \sigma(z^{i,convl}) \tag{11}$$

**b)** If *convl* is the convolution layer then,

$$\delta^{i,convll} = \delta^{i,convl} \times rot^{180} wt^{convl+1} \ominus \sigma(z^{i,convl}) \tag{12}$$

**c)** If *convl* is pool layer then,

$$a^{convl} = upsample(\delta^{i,convl+1}) \ominus \sigma(z^{i,cconvl}) \tag{13}$$

**Step 9:** End for

**Step 10:** To minimize the error rate, update the weight and bias

**Step 11:** *For cl = 2 to N − 1*

**a)** If *convl* is the fully connection layer then,

$$wt^{convl} = wt^{convl} - \alpha \sum_{i=1}^{m} \delta^{i,convl}(a^{i,convl-1})^{iter} \tag{14}$$

$$bi^{convl} = bi^{convl} - \alpha \sum_{i=1}^{m} \delta^{i,convl} \tag{15}$$

**b)** If *convl* is the convolution layer then,

$$wt^{convl} = wt^{convl} - \alpha \sum_{i=1}^{n} \delta^{i,convl} \times (a^{i,convl-1}) \tag{16}$$

$$bi^{convl} = bi^{convl} - \alpha \sum_{i=1}^{m} \sum_{\mu,v} \left( \overset{..i,convl}{a} \right)_{\mu,v} \tag{17}$$

**Step 12:** End for

**Step 13:** Terminate condition of DCNN using:

$$if \left( ||a^{it+1} - a^{it}|| < \varepsilon \text{ or } it < iter \text{ then loop ends} \right)$$

**Step 14:** Else go to step 1.

**Step 15:** **Implementing the Bagging process** by using:

*M* is the number of base classifiers and the classification label is defined as Q = {−1.+1}. The bagging method is declared in Eq. (18)

$$Q = G(x) = sign\left( \sum_{i=1}^{N} g_i(x) \right) \tag{18}$$

**Step 8: Output:** return *Q* and the relation coefficient matrix *wt* and *bi*.

Hence, the DCNN is used to select the optimal relevant features. To minimize the error rate and loss function, the Heuristic Multiswarm Ant colony Optimization (HMACO) is applied.

### Heuristic multiswarm ant colony optimization HMSACO

The HMSACO algorithm is implemented for the optimal detection of attacks in the network. The implementation process of HMSACO contains $CN = \{cn_1, cn_2, \ldots, cn_m\}$ set of components (nodes) or features and $E$ represents the collection of edges. A finite set of possible connections in the elements of $CN$ is defined by a subset of $CN$ of the Cartesian product of $CN \times CN$,

$$E = \left\{ \left(e_{cn_i,\, cn_j}\right) \in CN \right\}, E = \left\{ e_{cn_i,\, cn_j} | \left(cn_i,\, cn_j\right) \in CN \right\},\ |E| \leq M_{cn}^2 \tag{19}$$

For each $e_{cn_i,\, cn_j}$ is the cost function of connection between components in the period of time $t$. Applying the transition rule by using Eq. (20).

$$nM_{p,q}^l(s) = \frac{[\tau_{p,q}]^\alpha [\eta_{p,q}]^\beta}{\sum_{q \in Ra_i^l} [\tau_{p,q}]^\alpha [\eta_{p,q}]^\beta} \tag{20}$$

Equation (20) is implemented when $i \in CN_i(j)$

After implementing the transition rule in the components, the process's sequence is repeated until it satisfies the stop condition. During the repeating process, it updates the trials, evaluates the new solution, and remembers the optimal solution. Consider the graph $G = (CN, E)$. The feasible path of the graph is defined as $G$ for the optimized solution. The Ant Colony Algorithm (ACO) aims to find the minimum cost of sequences of the path with its feasible solution concerning its constraints. In ACO, the colony's population is considered agents or ants, which collectively identify the solution based on the graphical representation of the problem. The information gathered by the ants during its search process is defined as pheromone trails. The algorithmic procedure of HMSACO is given below:

**Input:** Optimal selection of relevant features
**Output:** Detection of attack $DA_{best}$
**Step 1:** $DA_{best} \leftarrow Generate\ Hwuristicsolution\ (Dataset)$
**Step 2:** Initialize the Pheromone with its parameters $\tau_{p,q}$.
**Step 3:** $DA_{best} \leftarrow cost\ (e_{cn})$
**Step 4:** While stop Iteration
**Step 5:** *For $(y = 1\ to\ parameters.\ M)$*
**Step 6:** $e_{cn}\_best \leftarrow buidsolution(Pheromone,\ graph,\ parameters)$
**Step 7:** If $e_{cnbest} \leq DA_{best}$ then $DA_{best} \leftarrow e_{cnbest}$
**Step 8:** End
**Step 9:** Local Update and $delay - Pheromone(Pheromone, e_{cnbest}\ parameters)$
**Step 10:** End
**Step 11:** Global update and $delay - Pheromone(Pheromone, DA_{best}\ parameters)$

**Step 12:** End

**Step 13:** Return the attack detection $DA_{best}$.

In the above procedure, ants start it process from the initial state and move towards the feasible node of their neighborhood states and build a solution in a forward movement of the ant. The constriction process gets stopped when it satisfies at least one of the termination conditions. Applying the transitive rule, an ant $l$ is start from node $m$ to node $n$ and it updates its pheromone trail $\tau_{p,q}$. This updating procedure is called a step-by-step pheromone. Once the solution is built, the ant has the capability to retrace the same path in the backward propagation method and updates its pheromone trails. This is called an online-delayed-pheromone

# RESULTS AND DISCUSSION

The proposed work DCNN-HMACO is used in the detection of attacks in the network using a cyber-physical system and the results are analyzed based on the performance metric measures of attack prediction ratio, accuracy, cost of communication, the ratio of delay, the ratio of efficiency, sensitivity, specificity, and F1-score. This dataset is split into 80% training and 20% testing dataset. This proposed work is compared with existing algorithms of CNN (*Thiruloga, Kukkala & Pasricha, 2022*), RNN (*Yoginath et al., 2019*).

## Dataset description

The datasets used in this work were UNSW-NB15 and TON_IoT Train_Test Network. which contain 2.55 million sample data. For this work, sample datasets were randomly chosen from UNSW-NB15, 150,282 for training and 27,895 for testing. Similarly, for TON_IoT Train_Test dataset, 160,576 samples were chosen for the training dataset and 30,602 samples for the testing dataset. Results were analyzed based on the performance metric measures of the Ratio of attack prediction, Accuracy detection, cost of communication, the ratio of delay, ratio of efficiency, sensitivity, specificity, and F1-score. Table 2 shows the features used in the UNSW-NB15 dataset.

## Ratio of attack detection

The attack detection of networks in cyber-physical systems is required because it can result in stealing the personal credentials of the user. Therefore, the proposed work DCNN-HMACO detects the attack in the network. Figure 5 demonstrates the detection of new attacks in the network of cyber-physical systems promptly detected by our proposed work of DCNN-HMACO and compared it with other existing algorithms of CNN and RNN.

Table 3 shows the TON_IoT Train_Test Network dataset

## Ratio of accuracy in detection of attack

Detection of attack in the CPS, in the aspect of accuracy rate, is implemented by using DCNN-HMACO. Figure 6 shows the accuracy detection of the attack. It demonstrates that the detection of attacks in the network of cyber-physical systems is high when implementing the DCNN-HMACO algorithm. It produces a high accuracy rate in detecting new arrival of aggression in the network.

**Table 2 Features of UNSW-NB15 dataset.**

**UNSW-NB15 dataset**

| Feature number | Feature name | Feature number | Feature name |
|---|---|---|---|
| f1 | dur | f23 | dwin |
| f2 | proto | f24 | tcprtt |
| f3 | service | f25 | synack |
| f4 | state | f26 | ackdat |
| f5 | spkts | f27 | smean |
| f6 | dpkts | f28 | dmean |
| f7 | sbytes | f29 | trans_depth |
| f8 | dbytes | f30 | response_body_len |
| f9 | rate | f31 | ct_srv_src |
| f10 | sttl | f32 | ct_state_ttl |
| f11 | dttl | f33 | ct_dst_ltm |
| f12 | sload | f34 | ct_src_dport_ltm |
| f13 | dload | f35 | ct_dst_sport_ltm |
| f14 | sloss | f36 | ct_dsc_src_ltm |
| f15 | dloss | f37 | is_ftp_login |
| f16 | sinpkt | f38 | ct_ftp_cmd |
| f17 | dinpkt | f39 | ct_flw_http_mthd |
| f18 | sjit | f40 | ct_src_ltm |
| f19 | djit | f41 | ct_srv_dst |
| f20 | swin | f42 | is_sm_ips_ports |
| f21 | stcpb | f43 | attack_cat |
| f22 | dtcpb | f44 | label |

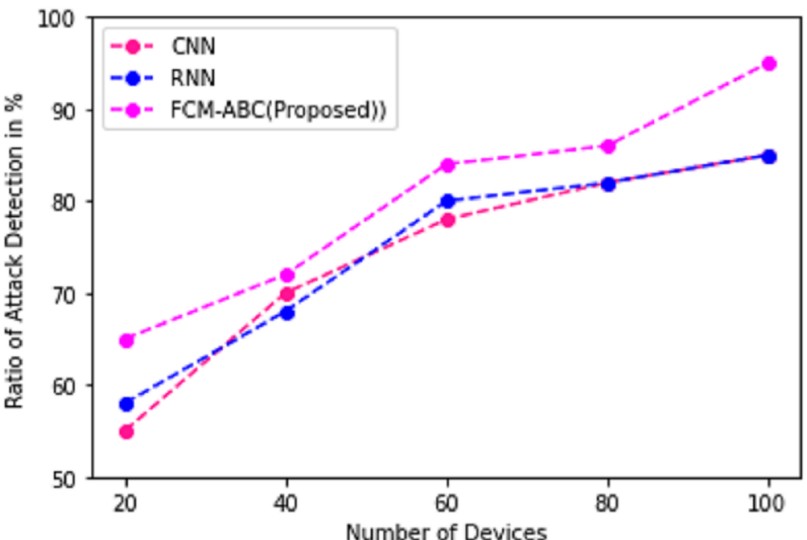

**Figure 5 Ratio of attack detection.**

**Table 3 Features of the TON_IoT Train_Test network dataset.**

**TON_IoT Train_Test network dataset**

| Feature number | Feature name | Feature number | Feature name |
|---|---|---|---|
| f1 | dns_AA | f24 | http_response_body_len |
| f2 | dns_RD | f25 | http_user_agent |
| f3 | dns_RA | f26 | label |
| f4 | dns_rejected | f27 | src_ip |
| f5 | ssl_resumed | f28 | dst_ip |
| f6 | ssl_established | f29 | proto |
| f7 | weird_notice | f30 | service |
| f8 | src_port | f31 | conn_state |
| f9 | dst_port | f32 | dns_query |
| f10 | duration | f33 | ssl_version |
| f11 | src_bytes | f34 | ssl_cipher |
| f12 | dst_bytes | f35 | ssl_subject |
| f13 | missed_bytes | f36 | ssl_issuer |
| f14 | src_pkts | f37 | http_method |
| f15 | src_ip_bytes | f38 | http_uri |
| f16 | dst_pkts | f39 | http_referrer |
| f17 | dst_ip_bytes | f40 | http_version |
| f18 | dns_qclass | f41 | http_orig_mime_types |
| f19 | dns_qtype | f42 | http_resp_mime_types |
| f20 | dns_code | f43 | weird_name |
| f21 | http_trans_depth | f44 | weird_addl |
| f22 | http_request_body_len | f45 | type |
| f23 | http_status_code | f46 | ts |

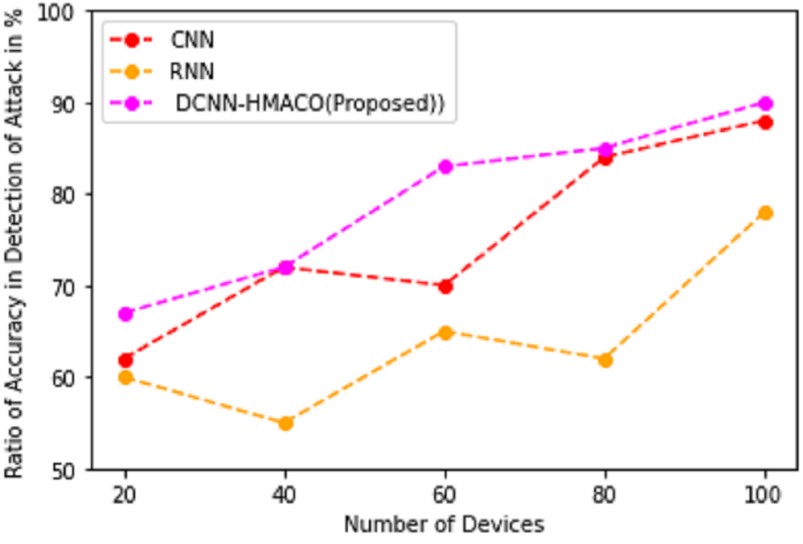

**Figure 6 Ratio of accuracy in detection of attack.**

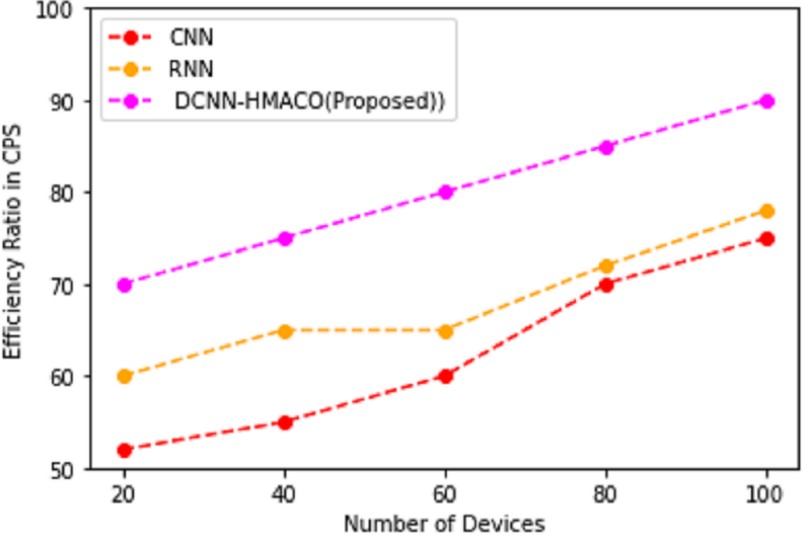

**Figure 7  Efficiency ratio in CPS.**                         

**Table 4  Record types in the ToN-IoT dataset.**

| Event type | Total records of data | Record of train-test |
|---|---|---|
| DDOS | 6,165,009 | 20,000 |
| Backdoor | 508,115 | 20,000 |
| Injection | 452, 658 | 20,000 |
| DoS | 3,375,327 | 20,000 |
| MITM | 1,053 | 1,042 |
| Ransome ware | 72,806 | 20,000 |
| Scanning | 7,140,162 | 20,000 |
| XSS | 2,108,943 | 20,000 |
| Password | 1,718,567 | 20,000 |
| Normal | 796,380 | 300,000 |
| Total | 22,339,018 | 461,042 |

## Efficiency ratio in CPS

Using the DCNN-HMACO algorithm detection of attack in the network of CPS based on features in the dataset of UNSW-NB15 and TON_IoT Train_Test Network dataset, Fig. 7 shows the efficiency ratio in CPS.

In the observation of Fig. 7, the efficiency ratio in cyber-physical systems for the detection of attack in the network is based on the terms of privacy-preserving of credentials information, secure in an efficient manner by using DCNN-HMACO. The performance of metric measures of:

$$Sensitivity = \frac{TP}{TP + FN} \tag{21}$$

**Table 5 Performance metric measures in the training dataset.**

| Algorithm | Training | | | | |
| --- | --- | --- | --- | --- | --- |
| | Sensitivity | Specificity | F1-score | Miss rate | Fall out |
| CNN | 0.75 | 0.82 | 0.79 | 0.088 | 0.075 |
| RNN | 0.85 | 0.75 | 0.73 | 0.081 | 0.068 |
| DCNN-HMACO | 0.91 | 0.93 | 0.92 | 0.052 | 0.043 |

**Table 6 The Win7, Win10, Network, Win10–Network datasets with numbers of normal records and the types of attachments collected.**

| Events | Win10 | Win7 | Win10-network | Network |
| --- | --- | --- | --- | --- |
| DDOS | 4,608 | 2,134 | 498,920 | 508,116 |
| backdoor | — | 1,179 | – | 508,116 |
| Injection | 612 | 998 | 24,311 | 452,658 |
| DoS | 525 | – | 109,957 | 3,375,328 |
| MITM | 15 | – | 87 | 1,052 |
| Ransome ware | – | 82 | – | 72,805 |
| Scanning | 447 | 226 | 208,572 | 7,140,161 |
| XSS | 1,269 | 4 | 106,746 | 21,089,844 |
| Password | 3,628 | 757 | 101,398 | 1,718,568 |
| Normal | 24,871 | 22,387 | 23,763 | 796,380 |

**Table 7 Performance metric measures in the testing dataset.**

| Algorithm | Testing | | | | |
| --- | --- | --- | --- | --- | --- |
| | Sensitivity | Specificity | F1-score | Miss rate | Fall out |
| CNN | 0.85 | 0.70 | 0.74 | 0.072 | 0.065 |
| RNN | 0.82 | 0.69 | 0.64 | 0.067 | 0.048 |
| DCNN-HMACO | 0.94 | 0.91 | 0.87 | 0.032 | 0.038 |

$$Specificty = \frac{TN}{TN + FP} \tag{22}$$

$$accuracy = \frac{TP + TN}{TP + TN + FP + FN} \tag{23}$$

$$miss\ rate(FPR) = \frac{FN}{TP + FN} \tag{24}$$

$$fall\ out(FNR) = \frac{FP}{TN + FP} \tag{25}$$

$$F - Score = 2 \times \frac{Precision \times Recall}{Precision + Recall} \tag{26}$$

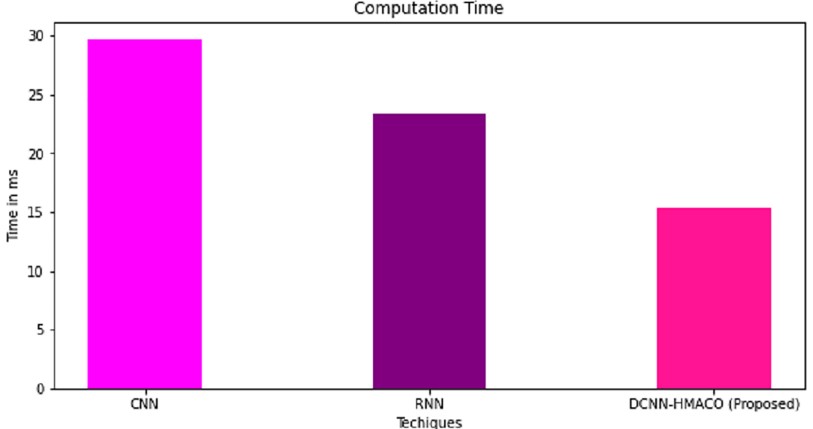

**Figure 8 Computation time.**                

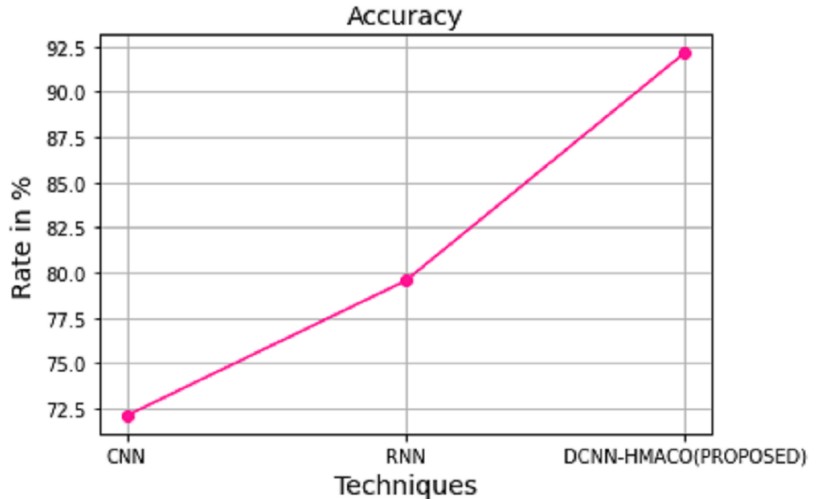

**Figure 9 Accuracy.**                

Table 4 shows Record Types in ToN-IoT dataset and Table 5 shows the performance metric measures of sensitivity, specificity, F1-Score, Miss Rate, Fall out in the training dataset.

Table 6 discussed the Win7, Win10, Network, and Win10–Network datasets with numbers of standard records and the types of attachments collected.

In the testing dataset, the proposed work DCNN-HMACO produces the metric measures of sensitivity of 0.94; specificity got 0.91, F1-Score got 0.87, miss rate got 0.032, and fall out of 0.038 is shown in Table 7.

The computation time and accuracy of DCNN-HMACO in detecting attacks in the network in CPS are compared with other algorithms like CNN and RNN. As shown in Figs. 8 and 9, respectively, the proposed work of DCNN-HMACO required minimum computation time and a higher attack detection accuracy rate.

## CONCLUSION

An emerging framework for attack detection in cyber-physical systems using the heuristic based optimization algorithm of DCNN-HMACO has been proposed. The dataset used in this work is UNSW-NB15 and TON_IoT Train_Test Network for the detection of attacks in the network.

To minimize the error loss function and enhance the detection of attacks in the network, the optimization algorithm is used. In the overall analysis of the detection of attack by using DCNN-HMACO in the aspects of accuracy, computation time, FPR, precision, recall, and F1 score, the accuracy rate of our proposed work got 92.14 % in comparison with other algorithms such as CNN that got 72.12% and FCM 79.56 %. In future work, the proposed DCNN-HMACO will be implemented with fuzzy-based detection of attack in the network.

### Funding

This work was supported by the Deanship for Research & Innovation, Ministry of Education in Saudi Arabia through the project number RI-44-0658. The funders had no role in study design, data collection and analysis, decision to publish, or preparation of the manuscript.

### Grant Disclosures

The following grant information was disclosed by the authors:
Deanship for Research & Innovation, Ministry of Education: RI-44-0658.

### Competing Interests

The authors declare that they have no competing interests.

### Author Contributions

- Manal Abdullah Alohali conceived and designed the experiments, performed the computation work, prepared figures and/or tables, and approved the final draft.
- Muna Elsadig performed the experiments, analyzed the data, prepared figures and/or tables, authored or reviewed drafts of the article, and approved the final draft.
- Anwer Mustafa Hilal performed the experiments, performed the computation work, authored or reviewed drafts of the article, and approved the final draft.
- Abdulwahed Mutwakel conceived and designed the experiments, analyzed the data, prepared figures and/or tables, authored or reviewed drafts of the article, and approved the final draft.

### Data Availability

The data is available at Kaggle:
- https://www.kaggle.com/datasets/mrwellsdavid/unsw-nb15.
- https://www.kaggle.com/datasets/dhoogla/nftoniot.

## Supplemental Information

Supplemental information for this article can be found online at http://dx.doi.org/10.7717/peerj-cs.1596#supplemental-information.

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
