# Peer review of "Emerging framework for attack detection in cyber-physical systems using heuristic-based optimization algorithm"

_PeerJ Computer Science, doi:10.7717/peerj-cs.1596_

## Round 0.1 · original submission · Major Revisions

Comments must be addressed to make the paper of very good standard.

Reviewer 1 ·

Basic reporting

The main focus of this paper is to introduce a framework for attack detection in Cyber-Physical Systems using Heuristic-Based Optimization Algorithm. The authors have utilized a deep bagging convolution neural network with Heuristic Multiswarm Ant Colony Optimization (DCNN-HMACO). The paper is written fairly. However, the current MS lacks in many aspects. Lacking concrete analysis and discussion of results and models. This research requires a lot more rigor.

Experimental design

In methodology, the authors should only focus on the adopted approach. It's not the section to define or provide the types of attacks. The attack types should be in the introduction section or in the Literature review with proper references.


More detail is needed in the Data collection subsection, and change its heading. Data collection only can be used if the authors have collected the data by themselves.

In lines 163 and 164, two consecutive headings create ambiguity.

The authors gained higher results as compared to the CNN and RNN, how the results were improved. There is a need to add critical analysis in the results section.

Validity of the findings

The results were compared with basic models CNN and RNN. However, many research articles on the same topic have achieved higher results, with more than 92% accuracy. Authors have to compare with such research works.
Most of the equations need references; not even a single equation is referred

UNSW 2 NB15 and TONIoT these datasets need a reference or a footnote to the address
The comparison performed with CNN and another algorithm, please mention if have performed by the authors or refer to someone else work
Most of the performance metrics discussed are related to classification tasks and how they will help in this ratio-related task.
Results comparison with CNN and other algorithms must indicate sample size comparison too.

Additional comments

2. Line 26, an unnecessary sentence,
3. Line 51-60 research objectives and contributions are mixed or used alternatively. These lines are misleading.
4. Review of Literature should be a literature review
5. Literature review is insufficient, authors failed to describe the current state of the research problem
6. Line 72, repeating words at the beginning of the line.
7. From the literature review, the reader may be confused about the research problem i.e., attack detection, prevention and resources protection or identification of cybercriminal attacks
8. Types of attacks in CPS may be described in the Introduction section or Background section instead of the Methodology section.
9. Table No. 1 survey on IDS needs its justification in the Methodology section; it is incomplete if it is a survey.

Reviewer 2 ·

Basic reporting

The introduction needs to discuss the research gaps well supported by up to date references. The current one is too short.
The contributions were poorly described.

"The main contribution of this work is as follows. 52 1. The main contribution of this work is the implementation of network attack detection in a cyber53 physical system using the pre-processing concept. 54 2. The main task described in this paragraph is extracting features from the dataset using a Linear 55 Discriminant algorithm. 56 3. The main objective is enhancing efficient attack detection using a deep convolutional neural network57 based bagging concept with Heuristic Multiswarm Ant Colony Optimization (DCNN-HMACO).”
2.1: using the pre-processing concept.? is PP is a concept?
2.3: the main task, the main contribution, the main objective? this lead to confusion. Please re-write the contributions well to describe the novelty of this research, the proposed a framework for detecting attacks, … etc.

3. Related studies are short too. It will be useful to add more recent studies in the table. And discuss them well in the text.

4. Add references (if any) to Figure 1. It will be helpful to re-size it to be smaller.

Experimental design

5. In methodology section, no need to review/define different types of attacks. You need to focus on the proposed framework. You may move these sections to related work.

6. Improve the quality of all figures. Example: Fig 3, Fig 5 &6, … etc.

7. Rename this section: Attack Detection using DCNN-HMACO (Proposed) Deep CNN

The proposed method: ….?

8. Move the evaluation metrics to the methodology.

9. Data description can be moved to the methodology. Add reference to this dataset.

Validity of the findings

Add more comparisons (with studies from LR) in the results section.

Discuss the impact of the pre-processing methods (before and after) in the results.

Enrich the results with more detailed discussion (and findings? )

Additional comments

Please consider the comments above.

·

Basic reporting

The proposed approach for network attack detection involves the application of DCNN with HMACO and LDA for feature extraction. The architecture of the DCNN consists of an input layer, convolution layer, pooling layer, fully connected layer, and output layer. The convolution layer extracts features from the dataset, which are then passed to the lower layer for further processing. An activation function is applied at this stage. The pooling layer performs sub-sampling to reduce the data obtained from the convolution layer. Nodes in the completely connected layer are connected to each and every node that existed in the layer below them. The softmax function is utilized on the output layer in order to identify malicious network activity.

Experimental design

No Comments

Validity of the findings

No Comments

Additional comments

The paper is well written and organized.

---

## Round 0.2 · accepted · Accept

The author has responded to all comments that have been provided by the reviewers.

Reviewer 1 ·

Basic reporting

The paper has been revised according to my previous suggestions. Therefore, I recommend it for acceptance

Experimental design

the experiments are now improved and satisfied

Validity of the findings

i am satisfied with the findings

Additional comments

No more comments from my side

Reviewer 2 ·

Basic reporting

The authors implemented the given corrections.

Experimental design

The experimental design was revised and improved.

Validity of the findings

The results were evaluated well.

·

Basic reporting

There is no comment; the author answered all the required questions.

Experimental design

The paper structured is very well.

Validity of the findings

I agree with his findings

Additional comments

No additional comments.